# Is brain imaging necessary for febrile elderly patients with altered mental status? A retrospective multicenter study

**Sungwoo Choi**[1⊙], **Hyun Na**[2⊙], **Sangun Nah**[1], **Hayeong Kang**[1], **Sangsoo Han**[1]*

**1** Department of Emergency Medicine, Soonchunhyang University Bucheon Hospital, Bucheon-si, Gyeonggi-do, Republic of Korea, **2** Department of Emergency Medicine, Andong General Hospital, Andong-si, Gyeongsangbuk-do, Republic of Korea

⊙ These authors contributed equally to this work.
* brayden0819@daum.net

## Abstract

### Objective

Altered mental status (AMS) is one of the most common symptoms in the febrile elderly. Brain imaging tests are an important tool for diagnosing AMS patients. However, these may be prescribed unnecessarily in emergency departments, particularly for febrile patients with AMS for whom infection is suspected, leading to excessive radiation risk and cost. In this study, we investigated the factors that can predict clinically significant abnormal brain imaging (ABI) in the febrile elderly with AMS.

### Methods

This retrospective multicenter study was conducted from July 2016 to June 2019. Febrile patients over the age of 65 years with AMS who visited the emergency department of two tertiary university hospitals were enrolled. Medical records were reviewed, and laboratory results were obtained. Brain imaging results with a formal reading by a radiologist were obtained.

### Results

In all, 285 patients were enrolled, and 47 (16.49%) showed ABI. The most common diagnoses in patients admitted to the emergency department were intracranial hemorrhage and ischemic stroke for ABI, and pneumonia and urinary tract infection for non-ABI. In multivariate logistic regression analyses, higher systolic blood pressure (odds ratio [OR], 1.017; 95% confidence interval [CI], 1.006–1.028), lower body temperature (OR, 0.578; 95% CI, 0.375–0.892), the presence of lateralizing sign (OR, 45.676; 95% CI, 5.015–416.025), and lower Glasgow Coma Scale (OR, 0.718; 95% CI, 0.617–0.837) were significantly associated with ABI.

**Data Availability Statement:** All relevant data are within the manuscript and its Supporting Information files.

**Funding:** This work was supported by the Soonchunhyang University Research Fund The funders had no role in study design, data collection and analysis, decision to publish, or preparation of the manuscript.

**Competing interests:** The authors declare that there are no conflicts of interest regarding the publication of this paper.

## Conclusion

Lower Glasgow Coma Scale, the presence of lateralizing sign, higher systolic blood pressure, and lower body temperature are significantly associated with ABI in febrile elderly patients with AMS.

## Introduction

The number of elderly people in the United States has rapidly increased from 25.5 million in 1980 to 49.2 million in 2016 [1]. As the proportion of the elderly population has increased, the medical utilization rate has also increased. In the United States, elderly patients using the emergency room increased from 12.4% in 2006 to 16% in 2016 [2]. In addition, the hospitalization rate of elderly patients is more than three times higher than that of non-elderly patients, and the hospitalization rate in intensive care units is five times higher [3, 4].

Altered mental status (AMS) is a common symptom in elderly patients, and the main causes are neurological (36.5%) and infectious (39.5%) [5]. Unlike non-elderly patients, elderly patients often experience decreased consciousness due to fever caused by viral or bacterial infections [5, 6]. Brain imaging is very helpful for identifying decreased consciousness due to neurological causes, such as cerebral infarction, but it is not useful when this is caused by infection [7, 8]. Therefore, it is not cost effective to perform brain imaging in all patients with AMS [9]. Still, most febrile elderly patients with AMS who have suspected infection have had unnecessary brain imaging tests. In addition, it may be risky for a patient with decreased consciousness to leave the emergency department for brain imaging and move to a radiology department [10].

Therefore, we studied the clinical factors that can predict abnormal brain imaging (ABI) in febrile elderly patients (over the age of 65 years) with AMS.

## Materials and methods

### Study population

This retrospective study was conducted on patients who visited the emergency center of two tertiary hospitals in Korea for three years from July 2016 to June 2019. Febrile patients over the age of 65 years who showed AMS were enrolled. Fever was defined as a case where the patient's body temperature (BT) was 38 degrees or more at the time of visit hospital or at home. AMS was defined as a state of Glasgow Coma Scale (GCS) of 14 or less, or new onset of altered consciousness such as drowsiness, behavior change, unresponsiveness, disorientation or confusion [9]. The patients who did not take a brain imaging study, such as brain computed tomography (CT) or brain magnetic resonance imaging (MRI) were excluded. Also, patients with history of head trauma, insufficient medical records and discharged against medical advice were excluded (Fig 1).

### Data collection

Medical records were reviewed for information on illness, examinations, vital signs, medications, and other data; we also obtained laboratory results including white blood cell count and levels of C-reactive protein, procalcitonin, and other parameters.

The onset of AMS was classified into acute (less than 24 h), subacute (1–7 days), and chronic (>7 days) [9]. Witnessed is defined as AMS that is seen or heard by another people

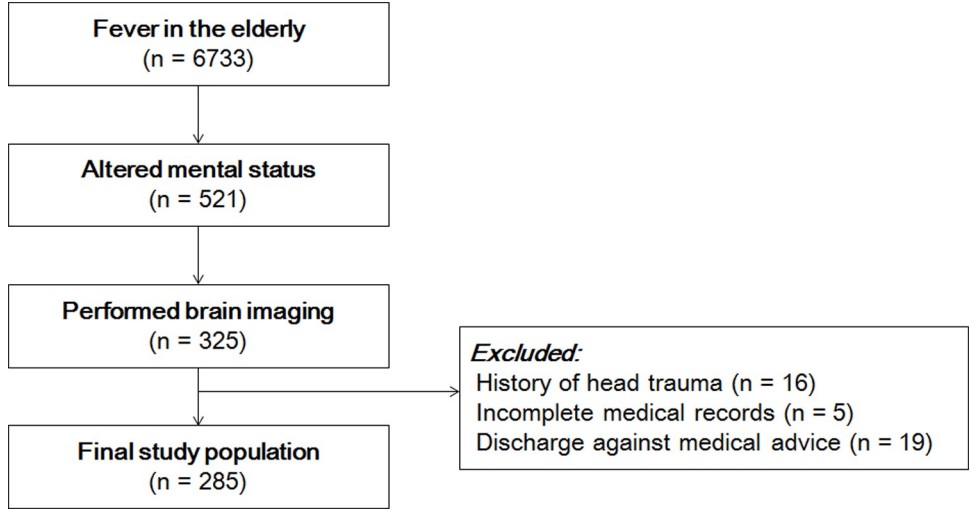

**Fig 1. Study population.**

when it happened [11]. All patients received physical and neurological examinations, conducted by a senior resident or emergency medicine specialist. Brain imaging was interpreted by a radiology specialist. All patient's medical records and results were reviewed by two emergency physicians, who had more than 10-years of clinical experience. If imaging results were associated with decreased consciousness, the case was defined as being clinically significant ABI. Otherwise, it was considered non-clinically significant ABI (NABI); chronic ischemic change, brain atrophy, and cerebromalacia were classified as NABI.

## Statistical analyses

Continuous variables were confirmed using the Shapiro–Wilk test and a histogram and analyzed using the Student's t-test and the Mann–Whitney U-test. Categorical variables were analyzed using the chi-square test or Fisher's exact test. The odds ratio (OR) and 95% confidence interval (CI) were calculated using a multivariate logistic model. A multivariate logistic regression model was created using the results of univariate analyses. In the multivariate logistic regression model, lateralizing sign, GCS, systolic blood pressure (SBP), and BT were used. The area under the receiver operating characteristic curve of the multivariable logistic regression model was 0.8122 (CI, 0.744–0.88). In addition, a nomogram was constructed with those four model variables to predict the probability of ABI through scoring. Statistical analyses were performed using the IBM SPSS Statistics ver. 26.0 software (IBM Corp., Armonk, NY, USA) and R ver. 3.5.3 software (R Foundation for Statistical Computing, Vienna, Austria).

## Ethics approval

The study protocol was approved by the Institutional Review Board of Soonchunhyang University Bucheon Hospital (IRB No. 2020-03-023). The study was conducted in accordance with the provisions of the Declaration of Helsinki.

## Results

In all, 6733 patients who visited the emergency center with fever were aged 65 years or older. Among them, 521 patients had decreased consciousness, and of these, 325 had brain imaging results. A total of 285 patients were included in the final analyses, having excluded those with a

history of head trauma or whose medical records did not include sufficient information about the final diagnosis or treatment (Fig 1).

The general characteristics of the patients are shown in Table 1. The average age of the patients was 78 years and male were 137 (48.07%). Patients of ABI were 47 (16.49%).

Statistical comparisons of the ABI and NABI groups are shown in Table 2. There were no significant differences between the two groups in age or sex. GCS, BT, and SBP were 8 (7–11), 38.3˚C (37.8–38.6), and 130 (110–158) in the ABI group and 11 (9–13), 38.5˚C (38–39.3), and 120 mmHg (100–140) in the NABI group. Overall, 8 (17.02%) patients in the ABI group and 1 (0.42%) in the NABI group had a lateralizing sign.

The final diagnoses for each group are shown in Table 3. The most common diagnose were brain hemorrhage (42.55%) and ischemic stroke (38.3%) in the ABI group and pneumonia (42.86%) and urinary tract infection (UTI; 19.75%) in the NABI group.

The results of the univariate and multivariate logistic regression analyses are shown in Table 4. Higher SBP, lower BT, the presence of lateralizing sign, and lower GCS were significantly associated with ABI.

A nomogram was constructed to predict the probability of ABI, as shown in Fig 2, using these four significant variables (SBP, BT, lateralizing sign, and GCS) as factors. On the nomogram, lateralizing sign is the most effective factor for prediction of ABI, followed by GCS, BT, and SBP. Each individual predictor matches a score on the point scale axis from 0 to 100, and ABI can be predicted by summing the scores of the individual factors; the total points indicate a probability. A calibration plot for this nomogram, showing the relationship between the predicted probability according to the nomogram and the calculated probability, is shown in Fig 3.

**Table 1. General characteristics of subjects.**

|  | Total (n = 285) |
|---|---|
| Age, years | 78 ± 7.21 |
| GCS | 10 (8, 12) |
| Male, n (%) | 137 (48.07) |
| Underlying conditions, n (%) |  |
| HTN | 168 (58.95) |
| DM | 119 (41.75) |
| Malignancy | 36 (12.63) |
| CKD | 16 (5.61) |
| Dementia | 81 (28.42) |
| CVA | 75 (26.32) |
| Onset of altered mental status, n (%) |  |
| Acute | 232 (81.40) |
| Subacute | 42 (14.74) |
| Chronic | 11 (3.86) |
| Witnessed, n (%) | 141 (49.82) |
| History of, n (%) |  |
| Anticoagulant use | 18 (6.32) |
| Antiplatelet use | 69 (24.21) |
| ABI, n (%) | 47 (16.49) |

GCS: Glasgow Coma Scale, HTN: hypertension, DM: diabetes mellitus, CKD: chronic kidney disease, CVA: cerebrovascular accident, ABI: clinically significant abnormal brain imaging.

**Table 2. Comparison of clinical characteristics.**

| Characteristics | ABI (n = 47) | NABI (n = 238) | p-value |
|---|---|---|---|
| Age, years | 77 (69–83) | 78 (74–83) | 0.125 |
| Male, n (%) | 21 (44.68%) | 116 (48.74%) | 0.727 |
| GCS | 8 (7–11) | 11 (9–13) | < 0.001 |
| Vital signs | | | |
| SBP, mmHg | 130 (110–158) | 120 (100–140) | 0.015 |
| DBP, mmHg | 80 (70–90) | 80 (60–90) | 0.112 |
| Heart rate, /min | 92 (82–110) | 100 (89–110) | 0.255 |
| Respiratory rate, /min | 20 (20–24) | 22 (20–25) | 0.045 |
| Body temperature, ˚C | 38.3 (37.8–38.6) | 38.5 (38–39.3) | 0.003 |
| Onset of alerted mental status, n (%) | | | 0.174 |
| Acute | 34 (72.34) | 198 (83.19) | |
| Subacute | 10 (21.28) | 32 (13.45) | |
| Chronic | 3 (6.38) | 8 (3.36) | |
| Witnessed, n (%) | 18 (38.3) | 123 (52.12) | 0.116 |
| History of, n (%) | | | |
| Anticoagulant use | 4 (8.51) | 14 (5.88) | 0.511 |
| Antiplatelet use | 9 (19.15) | 60 (25.21) | 0.484 |
| Underlying conditions, n (%) | | | |
| HTN | 26 (55.32) | 142 (59.66) | 0.696 |
| DM | 20 (42.55) | 99 (41.6) | >0.99 |
| Malignancy | 10 (21.28) | 26 (10.92) | 0.087 |
| CKD | 2 (4.26) | 14 (5.88) | >0.99 |
| Dementia | 12 (25.53) | 69 (28.99) | 0.761 |
| CVA | 9 (19.15) | 66 (27.73) | 0.299 |
| Lateralizing sign, n (%) | 8 (17.02) | 1 (0.42) | < 0.001 |
| Laboratory test | | | |
| WBC, $10^3/\mu L$ | 13.41 (10.93–15.5) | 10.98 (7.21–15.36) | 0.047 |
| CRP, mg/L | 4.19 (0.61–14.61) | 7.12 (2.64–14.8) | 0.056 |
| Procalcitonin, ng/mL | 0.5 (0.45–0.98) | 1.1 (0.5–6.55) | 0.048 |
| Ammonia, µg/dL | 43 (31.75–57.75) | 43 (35–58.25) | 0.532 |
| Glucose, mg/dL | 161 (139.5–190) | 156.5 (118.25–219.5) | 0.246 |
| BUN, mg/dL | 21.6 (15.3–30) | 24.15 (17.02–38) | 0.092 |
| Creatinine, mg/dL | 1 (0.84–1.47) | 1.3 (0.9–1.9) | 0.035 |
| AST, IU/L | 32 (24.5–48) | 30 (20–53.75) | 0.308 |
| ALT, IU/L | 17 (12.5–26.5) | 19 (11–33.75) | 0.924 |
| CK, IU/L | 157 (80–366) | 94.5 (45.25–233) | 0.025 |
| Lactate, mmol/L | 4.8 (2.45–14.72) | 9.35 (7.41–7.5) | 0.219 |

ABI: clinically significant abnormal brain imaging, NABI: either normal or non-clinically significant abnormal brain imaging, GCS: Glasgow coma scale, SBP: systolic blood pressure, DBP: diastolic blood pressure, HTN: hypertension, DM: diabetic mellitus, CKD: chronic kidney disease, CVA: cerebrovascular accident, WBC: white blood cell, CRP: C-reactive protein, BUN: blood urea nitrogen, AST: aspartate aminotransferase, ALT: alanine aminotransferase, CK: creatine kinase.

## Discussion

For patients with AMS, brain imaging tests are an important diagnostic tool. However, elderly patients tend to show AMS when they have an infection or inflammation compared to non-

**Table 3. Final consensus diagnoses of altered mental status in febrile elderly patients.**

| Final diagnosis | |
|---|---|
| ABI, n (%) | |
| Brain hemorrhage | 20 (42.55) |
| Ischemic stroke | 18 (38.3) |
| Malignancy | 6 (12.77) |
| Others | 3 (6.38) |
| NABI, n (%) | |
| Pneumonia | 102 (42.86) |
| Urinary tract infection | 47 (19.75) |
| Metabolic encephalopathy | 21 (8.82) |
| Biliary tract infection | 11 (4.62) |
| Enterocolitis | 7 (2.94) |
| Influenza virus infection | 6 (2.52) |
| Liver abscess | 4 (1.68) |
| Heat stroke | 4 (1.68) |
| CNS infection | 3 (1.26) |
| Others | 33 (13.87) |

ABI: clinically significant abnormal brain imaging, NABI: either normal or non-clinically significant abnormal brain imaging, CNS: central nervous system.

elderly patients. It is thus challenging to identify the need for a brain imaging test when a febrile elderly patient, for whom infection is suspected, shows AMS. In this study, we identified four factors that predict ABI in such patients: higher SBP, lower BT, the presence of lateralizing sign, and lower GCS.

Studies have reported conflicting results in terms of abnormal CT findings and AMS in the elderly [12, 13]. In our study, 16.49% of patients had ABI, similar to one previous study [13]. The slight difference could be attributed to differences in enrollment criteria: we included febrile patients down to a slightly lower age (65 years) and also included MRI for detecting ABI. Another study reported that hemorrhage, infarction, and tumor were the most common final diagnoses [14]. Our results are in agreement with that study. In another study that analyzed patients aged over 65 who came to a hospital due to consciousness change, the most common cause was infection (39.5%), and pneumonia and UTI were the two most common diagnoses in this group [5]. We found similar results, with pneumonia and UTI as the two most common diagnoses in the NABI group. Therefore, in febrile elderly patients with AMS, it is necessary to check not only brain lesions but also diseases caused by infections.

To address AMS, it is very important to differentiate the pathologic problems of the brain in elderly patients who come to a hospital. Brain imaging tests are widely used diagnostic methods for patients who experience mental changes [15]. However, their use raises radiation risk and patient cost [14, 16]. In one study, the utilization rate of brain CT increased by 60% over a 7-year period while the number of brain hemorrhage diagnosed stayed constant around approximately 3% [10].

Therefore, it is important to find appropriate clinical factors to identify AMS patients who truly need brain tests. Lim *et al*. [17] presented seven clinical predictors (age, drowsiness or unresponsiveness, previous cerebrovascular accident or epilepsy, tachycardia, bradycardia, and exposure to drugs) of abnormal CT findings in patients with AMS. Shin *et al*. [14] concluded that a focal neurological deficit or a GCS < 9 and C-reactive protein < 2 mg/dL are

**Table 4. Adjusted odds ratio of exploratory variables associated with clinically significant abnormal brain imaging following univariate and multivariate logistic regression.**

| Variables | Univariate | | Multivariate | |
|---|---|---|---|---|
| | Odds ratio (95% CI) | p-value | Odds ratio (95% CI) | p-value |
| RR, /min | 0.932 (0.866–1.003) | 0.061 | | |
| WBC, $10^3/\mu L$ | 1.010 (0.981–1.041) | 0.494 | | |
| Procalcitonin, ng/mL | 0.959 (0.906–1.016) | 0.156 | | |
| Creatinine, mg/dL | 0.719 (0.498–1.040) | 0.079 | | |
| CK, IU/L | 1.000 (0.999–1.000) | 0.936 | | |
| Lactate, mmol/L | 0.988 (0.966–1.010) | 0.271 | | |
| SBP, mmHg | 1.013 (1.003–1.023) | 0.008 | 1.017 (1.006–1.028) | 0.002 |
| BT, ˚C | 0.592 (0.415–0.846) | 0.004 | 0.578 (0.375–0.892) | 0.013 |
| Lateralizing sign | 48.615 (5.916–3999.476) | <0.001 | 45.676 (5.015–416.025) | 0.001 |
| GCS | 0.719 (0.627–0.824) | <0.001 | 0.718 (0.617–0.837) | <0.001 |

ABI: clinically significant abnormal brain imaging, CI: confidence interval, RR: respiratory rate, WBC: white blood cell, CK: creatine kinase, SBP: systolic blood pressure, BT: body temperature, GCS: Glasgow Coma Scale.

closely related to abnormal findings in brain CT. In line with that study, we found that a lateralizing sign and GCS were predictive factors. In addition, SBP was a factor; indeed, high blood pressure is a risk factor for spontaneous intracranial hemorrhage, and uncontrolled

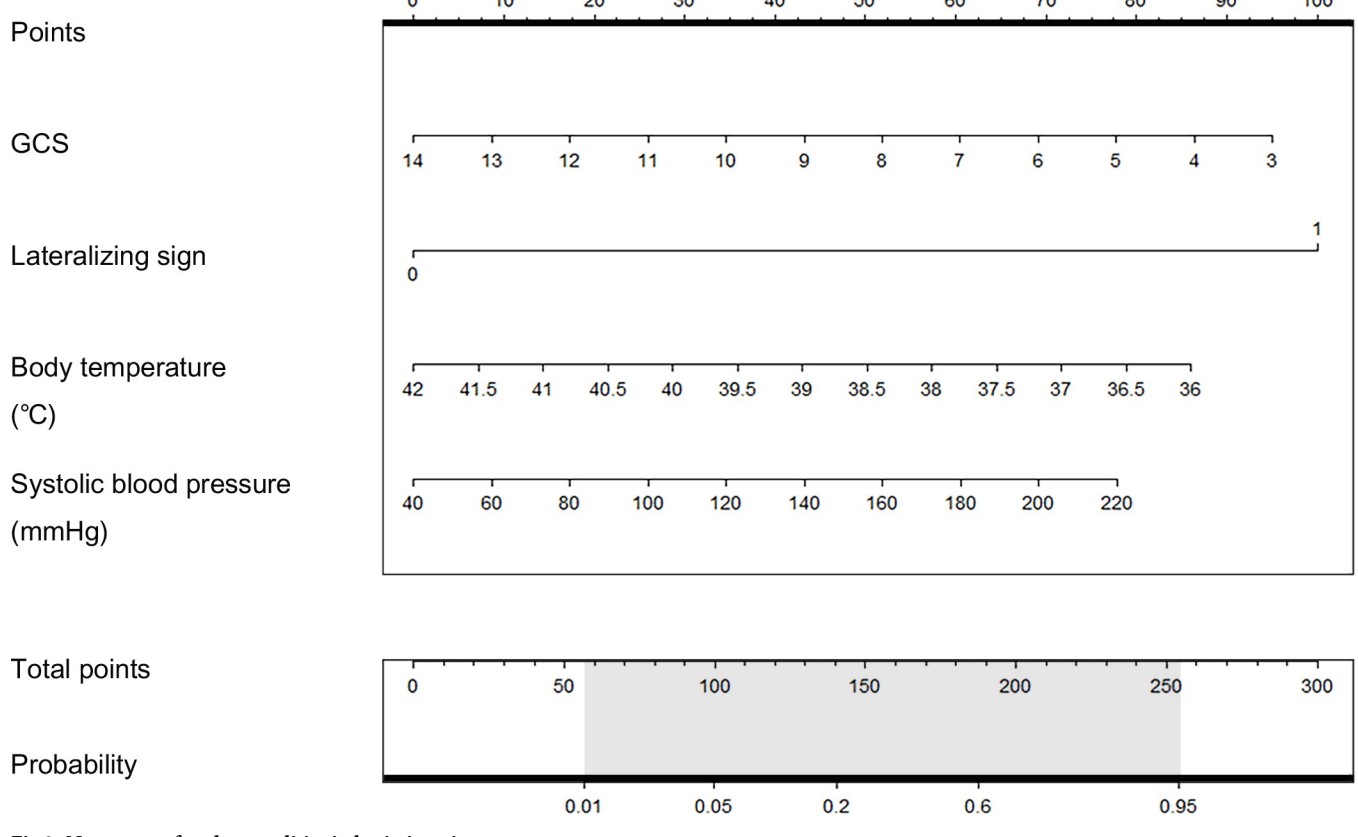

**Fig 2. Nomogram for abnormalities in brain imaging.**

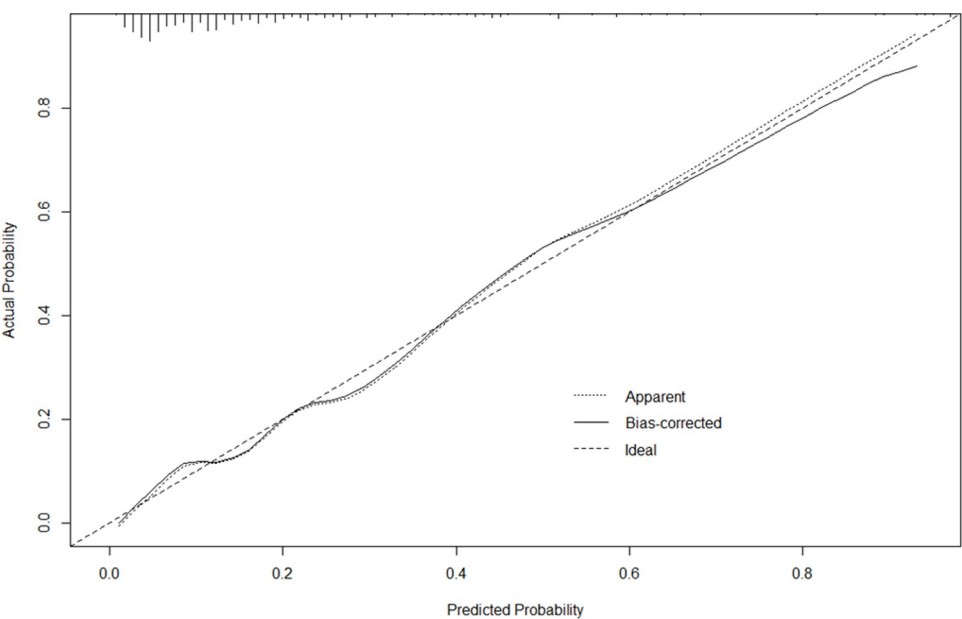

**Fig 3. The calibration plot of nomogram.**

hypertension can cause recurrence or re-bleeding [18, 19]. Finally, BT was also a predictive factor; indeed, high BT in an elderly person suggests that there is a serious viral or bacterial infection [20].

In the nomogram created to help decision-making (Fig 2), it can be seen that lateralizing sign occupies the most effective factor to predict ABI. Each factor is scored, and the probability is indicated, thus providing a meaningful decision-making tool for predicting brain anomalies and whether to proceed with examination by a clinician. These clinical findings can be an important measure for judging whether a test should be performed on an unconscious elderly patient with fever in a hospital. However, to generalize this, external validation is required.

There were several limitations to this study. First, because it was a retrospective study, selection bias may have occurred. Clinicians decide whether to perform brain imaging, and it may not be performed based on a patient's condition. In addition, research conducted at two hospitals may differ in their protocols. Second, we could not assess any changes in the degree of consciousness while patients were hospitalized; we considered only their initial consciousness at the time of their visit. Third, we included the cases who had fever at home in the study population [21, 22], but fever measured at home may not be reliable. Fourth, this study was conducted at two tertiary university hospitals in a large city, and it is unclear whether similar results would be obtained in other provinces or countries. Therefore, there is a limit to this study's generalizability, and there is a need for a prospective large-scale study.

## Conclusions

In elderly people, not only intracranial lesions but also inflammation or infection can easily result in AMS. Therefore, it is difficult to decide whether to conduct a brain test when a febrile elderly patient with AMS comes to a hospital. In this study, lower GCS, the presence of lateralizing sign, higher SBP, and lower BT were factors that predicted ABI in such patients.

## Supporting information

**S1 File. The datasets for statistical analysis.**
(XLSX)

## Author Contributions

**Conceptualization:** Hyun Na, Sangsoo Han.

**Data curation:** Sungwoo Choi, Hyun Na, Sangun Nah.

**Funding acquisition:** Sangsoo Han.

**Methodology:** Sungwoo Choi, Hayeong Kang.

**Project administration:** Sangsoo Han.

**Resources:** Sangsoo Han.

**Supervision:** Sangsoo Han.

**Writing – original draft:** Sungwoo Choi.

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
