## [Decision Letter · Decision Letter 0]

23 Jun 2020

PONE-D-20-15072

Is brain imaging necessary for febrile elderly patients with altered mental status? A retrospective multicenter study

PLOS ONE

Dear Dr. Han,

Thank you for submitting your manuscript to PLOS ONE. After careful consideration, we feel that it has merit but does not fully meet PLOS ONE’s publication criteria as it currently stands. Therefore, we invite you to submit a revised version of the manuscript that addresses the points raised during the review process.

We look forward to receiving your revised manuscript.

Kind regards,

Juan Manuel Marquez-Romero, M.D., M.Sc.

Academic Editor

PLOS ONE

Journal Requirements:

Reviewers' comments:

Reviewer's Responses to Questions

**Comments to the Author**

1. Is the manuscript technically sound, and do the data support the conclusions?

Reviewer #1: Partly

2. Has the statistical analysis been performed appropriately and rigorously? 

Reviewer #1: Yes

3. Have the authors made all data underlying the findings in their manuscript fully available?

Reviewer #1: Yes

4. Is the manuscript presented in an intelligible fashion and written in standard English?

Reviewer #1: No

5. Review Comments to the Author

Reviewer #1: This paper was related with brain imaging in elderly febrile patients with AMS. Some issues should be clarified in the study.

Abstract:

• What does it mean “Pneumonia and urinary tract infection were the most common in controls.” Authors wanted to mean that “The most common diagnoses in these patients admitted to the emergency department were pneumonia and urinary tract infection.”

• Authors should state “OR” and “95% CI” in the result when specifying logistic regression analysis.

Materials and Methods:

• AMS definition was poor, so authors should elaborate the AMS criteria.

• In data collection, authors stated that “Two emergency doctors reviewed the results”. This is one of the most important places in this study. How competent were the two doctors? Were the AMS diagnoses clear in all patients? These and similar issues should be more detailed.

• It was stated that “Fever was defined as a case where the patient's body temperature (BT) was 38 degrees or more at the time of visit hospital or at home.” Can home fever measurement be reliable for a retrospective study?

Result:

• In table 1, what does “witness” mean? It should be stated in the method or elsewhere.

6. PLOS authors have the option to publish the peer review history of their article (what does this mean?). If published, this will include your full peer review and any attached files.

Reviewer #1: Yes: Mehmet Ali Aslaner

---

## [Author Response · Author response to Decision Letter 0]

30 Jun 2020

Thank you for your e-mail concerning the further revision of our original manuscript entitled “Is brain imaging necessary for febrile elderly patients with altered mental status? a retrospective multicenter study”. We would like to thank both the editor and the reviewers for the constructive comments, which have helped us to improve the quality of the manuscript. These suggestions were very helpful and have significantly improved our manuscript.

---

## [Decision Letter · Decision Letter 1]

14 Jul 2020

Is brain imaging necessary for febrile elderly patients with altered mental status? A retrospective multicenter study

PONE-D-20-15072R1

Dear Dr. Han,

We’re pleased to inform you that your manuscript has been judged scientifically suitable for publication and will be formally accepted for publication once it meets all outstanding technical requirements.

Kind regards,

Juan Manuel Marquez-Romero, M.D., M.Sc.

Academic Editor

PLOS ONE

Additional Editor Comments (optional):

Reviewers' comments:

Reviewer's Responses to Questions

**Comments to the Author**

1. If the authors have adequately addressed your comments raised in a previous round of review and you feel that this manuscript is now acceptable for publication, you may indicate that here to bypass the “Comments to the Author” section, enter your conflict of interest statement in the “Confidential to Editor” section, and submit your "Accept" recommendation.

Reviewer #1: All comments have been addressed

2. Is the manuscript technically sound, and do the data support the conclusions?

Reviewer #1: Yes

3. Has the statistical analysis been performed appropriately and rigorously? 

Reviewer #1: Yes

4. Have the authors made all data underlying the findings in their manuscript fully available?

Reviewer #1: Yes

5. Is the manuscript presented in an intelligible fashion and written in standard English?

Reviewer #1: Yes

6. Review Comments to the Author

Reviewer #1: (No Response)

7. PLOS authors have the option to publish the peer review history of their article (what does this mean?). If published, this will include your full peer review and any attached files.

Reviewer #1: **Yes: **Mehmet Ali Aslaner

---

## [Editor Report · Acceptance letter]

17 Jul 2020

PONE-D-20-15072R1 

Is brain imaging necessary for febrile elderly patients with altered mental status? A retrospective multicenter study 

Dear Dr. Han:

I'm pleased to inform you that your manuscript has been deemed suitable for publication in PLOS ONE. Congratulations! Your manuscript is now with our production department. 

Kind regards, 

on behalf of

Dr. Juan Manuel Marquez-Romero 

Academic Editor

PLOS ONE